# Analysis of Long Non-Coding RNAs and mRNAs Associated with Lactation in the Crop of Pigeons (*Columba livia*)

**DOI:** 10.3390/genes11020201

**Published:** 2020-02-17

**Authors:** Hui Ma, Aixin Ni, Pingzhuang Ge, Yunlei Li, Lei Shi, Panlin Wang, Jing Fan, Adamu Mani Isa, Yanyan Sun, Jilan Chen

**Affiliations:** Key Laboratory of Animal Genetics Breeding and Reproduction (Poultry), Ministry of Agriculture, Institute of Animal Science, Chinese Academy of Agricultural Sciences, Beijing 100193, China; caumah@163.com (H.M.); naixin951@163.com (A.N.); gepingzhuang@163.com (P.G.); mailyunlei@163.com (Y.L.); shilei2017@foxmail.com (L.S.); wangpanlin0910@163.com (P.W.); fanjing9511@163.com (J.F.); 2018y90100036@caas.cn (A.M.I.); yanyansun2014@163.com (Y.S.)

**Keywords:** pigeon, crop, milk, lactation, lncRNA

## Abstract

Pigeons have the ability to produce milk and feed their squabs. The genetic mechanisms underlying milk production in the crops of ’lactating’ pigeons are not fully understood. In this study, RNA sequencing was employed to profile the transcriptome of lncRNA and mRNA in lactating and non-‘lactating’ pigeon crops. We identified 7066 known and 17,085 novel lncRNAs. Of these lncRNAs, 6166 were differentially expressed. Among the 15,138 mRNAs detected, 6483 were differentially expressed, including many predominant genes with known functions in the milk production of mammals. A GO annotation analysis revealed that these genes were significantly enriched in 55, 65, and 30 pathways of biological processes, cellular components, and molecular functions, respectively. A KEGG pathway enrichment analysis revealed that 12 pathways (involving 544 genes), including the biosynthesis of amino acids, the propanoate metabolism, the carbon metabolism and the cell cycle, were significantly enriched. The results provide fundamental evidence for the better understanding of lncRNAs’ and differentially expressed genes’ (DEGs) regulatory role in the molecular pathways governing milk production in pigeon crops. To our knowledge, this is the first genome-wide investigation of the lncRNAs in pigeon crop associated with milk production. This study provided valuable resources for differentially expressed lncRNAs and mRNAs, improving our understanding of the molecular mechanism of pigeon milk production.

## 1. Introduction

Pigeons (*Columba livia*) were domesticated in the Middle East during the Neolithic times and there are over 350 established domestic breeds [1]. The species is unique among birds as it can produce crop milk during the process of ‘lactation’ to nourish newly hatched squabs [2]. The first report of pigeon lactation in the literature was in 1786 by John Hunter, who described the crop milk as being ‘granulated white curd’ [3]. Both male and female pigeons possess the ability to synthesize crop milk, similarly to parent flamingos and male emperor penguins [4,5]. For other bird species, the crop only acts as a storage tissue located between the esophagus and the proventriculus, where food is moistened, intenerated and fermented.

The ‘lactating’ pigeon crop is anatomically different to the non-‘lactating’ crop, both macroscopically and histologically [2]. Many studies have investigated the genes involved in lactation. Horseman and Pukac first identified changes in gene expression by injecting prolactin into the crop [6]. The annexin Icp35 (*anxIcp35*) mRNA was reported to be the most abundant of the prolactin-induced genes expressed in the crop, which leds to the synthesis of the cp35 isoform of the annexin I protein [7]. Gillespie et al. probed differences in the gene expression in pigeon crops using chicken microarrays for the first time, and detected 542 more up-regulated genes and 639 more down-regulated genes in lactating crops than in non-lactating crops, by global gene expression profiling [2]. Gillespie then showed that pigeon milk production involved a specialized cornification process and a triglyceride synthesis in which the genes associated with cornification, including *β-keratin*, *S100-A9* and *S100-A10*, were up-regulated in the lactating crop [8].

Shapiro et al. submitted the pigeon reference genome and annotated sequences for the first time in 2013 by de novo genome assembly from sequences generated with the Illumina HiSeq2000 platform [1]. Damas combined computational algorithms to merge PCR-based scaffold verification into chromosomal fragments and improved the pigeon genome Scaffold N50 from 3.15 Mb to 22.17 Mb [9]. Recently, Holt et al. improved the reference genome assembly of pigeon, bringing the number of scaffolds to 15,057 with a N50 scaffold length of 14.3 Mb [10]. Although the transcriptome of the pigeon has been analyzed using microarrays in past years [2], next-generation sequencing technology has been widely used for transcript sequencing in recent years. All these progresses in pigeon genome studies have facilitated the analyses of the mechanism of pigeon physiology, including milk production.

Long non-coding RNAs (lncRNAs), a class of transcripts composed of at least 200 nucleotides that lack functional open reading frames, exert key roles by regulating gene expression in diverse biological processes [11]. Many studies have demonstrated that lncRNAs are involved in the lactation of the mammary gland by targeting and regulating protein-coding genes [12,13,14]. In this study, we employed high-throughput RNA-seq to analyze the gene expression profiles of lactating and non-lactating crops to identify the key lncRNAs and genes associated with crop milk production. The expression profile of the lncRNA in the lactating pigeon crop and its potential functions in milk production are still unknown. Studies of the candidate lncRNAs may improve the understanding of crop developmental processes and crop milk production.

## 2. Materials and Methods

### 2.1. Ethics Statement

All the work using animals was approved by the Animal Care and Use Committee at the Institute of Animal Sciences of the Chinese Academy of Agricultural Sciences (IAS 2018-3). All procedures were conducted in accordance with the institutional animal ethics guidelines set by the Ministry of Agriculture and Rural Affairs of the People’s Republic of China.

### 2.2. Sample Collection

A total of ten female White King pigeons of the same age were selected from a pigeon breeding farm in Beijing, comprising of five non-lactating pigeons and five lactating pigeons, 72 h after the hatching of their first squabs. The tissues were frozen rapidly in liquid nitrogen after the pigeons were slaughtered by CO_2_ asphyxiation.

### 2.3. Total RNA Extraction

The total RNA was extracted from samples of the crop tissue using TRIzol reagent (Invitrogen, USA) according to the manufacturer’s protocol. The RNA purity was checked using a Nanodrop Spectrophotometer (Model kaiaoK5500^®^, Beijing, China). The integrity and concentration of the RNA were measured by the RNA Nano 6000 Assay Kit of the Bioanalyzer 2100 system (Agilent Technologies, CA, USA).

### 2.4. Library Preparation for lncRNA Sequencing

Ten libraries ( ‘lactation’ = 5 libraries, non-‘lactation’ = 5 libraries) were constructed from 3 μg of total RNA for each sample using the NEB Next Ultra Directional RNA LibraryPrep Kit for Illumina (NEB, Ispawich, USA) according to the manufacturer’s instructions. Prior to the generation of the libraries, the rRNAs were removed using Ribo-zero GoldKits reagents. The sequencing was performed on an Illumina HiSeq 2500 instrument using the TruSeq PE Cluster Kit v4-cBot-HS to generate 150 bp paired-end reads.

### 2.5. Quality Control, Mapping, and Assembly of Transcriptomic Data

The raw reads generated from the ten libraries were filtered using FastQC (0.11.2). Raw data of RNA sequencing in this study was submitted to the Genome Sequence Archive (GSA) with accession no. CRA001977. The low-quality reads, the adaptor-polluted reads and the reads with >5% poly-N were discarded in the initial filtering step. The clean reads were mapped onto the rock pigeon colLiv2 reference genome sequence using HISAT2 (version 2.0.5, parameter --rna-strandness RF --dta–t–p 4) [15]. The mapped reads were assembled and merged into transcripts using StringTie software. StringTie (Version 1.3.2d) was run with the parameters -G ref.gtf–rf -l to build consensus sets of transcripts.

### 2.6. Identification of lncRNAs and Target Gene Prediction

The assembled transcripts were further subjected to a stringent filtering progress to obtain the candidate lncRNAs. First, we removed the transcripts with one exon and with lengths of less than 200 bp. Next, we calculated the reads’ coverage of transcripts using StringTie and removed transcripts with a reads’ coverage of less than three. Subsequently, we eliminated known genes, pre-micro RNA and other non-coding RNAs (rRNA, tRNA, snoRNA and snRNA) using Cuffcompare. Finally, the protein coding potency of the transcripts was calculated by using four software packages: the Coding-Non-Coding-Index (CNCI) (score < 0), the Coding Potential Calculator (CPC) (score < 0), the Pfam-scan (E-value < 0.001), and the Coding Potential Assessment Tool (CPAT) (score < 0). After filtering out the transcripts identified as having coding potential, transcripts that passed all the filters mentioned above were considered as candidate lncRNA. The candidate lncRNAs were then blasted to the ALDB database to classify the known and novel lncRNAs. We selected the mRNAs with high Spearman correlation coefficients (*p* ≥ 0.9) as the trans-targets and mRNAs with distances of less than 50 kb as the cis-targets.

### 2.7. Differential Expression and Functional Enrichment Analysis

The gene expressions were normalized using the fragments per kilobase of exon per million reads mapped (FPKM) method. The differentially expressed mRNAs and lncRNAs were analyzed using the R package DESeq (version 1.16.0, method = ‘per-condition’) based on negative binomial distribution. The multiple hypothesis testing correction of the *p*-value was adjusted using the Benjamini–Hochberg algorithm. A Q value < 0.05 and|log_2_foldchange|≥ 1 were considered as the significance threshold.

A GO term and KEGG pathway enrichment analysis was conducted using the DAVID program (v6.8) online tool [16]. An official gene ID was used and *Columba livia* was on the background list. The GO terms and KEGG pathways with FDR <0.05 were considered to be significantly enriched.

### 2.8. qRT-PCR Assay

The RNA samples were reverse transcribed into cDNA using the PrimeScript RT Reagent Kit (TaKaRa, Japan) following the manufacturer’s instructions. The concentration of the RNA samples was adjusted to 200 ng/μL by adding RNase free dH_2_O. The first step of the reverse transcription was conducted at 42 °C for 2 min. It contained 5 μL RNA samples, 1 μL gDNA Eraser, 2 μL 5 × gDNA Eraser Buffer and 2 μL RNase free dH_2_O. The second step was conducted at 37 °C for 15 min, then at 85 °C for 5 s. It contained 10 μL last step reaction solution, 1 μL PrimeScript RT Enzyme Mix, 4 μL 5 × PrimeScript Buffer, 1 μL RT Primer Mix and 4 μL RNase free dH_2_O.

A qRT-PCR was performed using the PrimeScript One Step RT-PCR Kit (TaKaRa, Japan) and the ABI QuantStudio 7 Flex Real-Time Detection System (Life Technologies Holdings Pte Ltd., USA). Each 10 μL qRT-PCR mixture contained 5 μL SYBR Premix Ex Taq^TM^ II, 0.5 μL (10 pM), and in each primer there was 0.2 μL ROX Reference Dye II (50×), 1.5 μL cDNA (100 ng) and 2.3 μL RNase free dH_2_O. After an initial denaturing at 95 °C for 3 min, there were 40 cycles of amplification (95 °C for 30 s and 60 °C for 34 s), followed by a thermal denaturing (95 °C for 15 s, 60 °C for 60 s, and 95 °C for 15 s) to generate melting curves. The primers of the genes and lncRNAs as shown in Appendix A were designed using the Primer Premier 5 and confirmed by Oligo 6.0. The *β-actin* was amplified in the same plates as an endogenous control and the relative-expression levels of genes and lncRNAs were quantified using 2^−ΔΔCt^ methods.

## 3. Results

### 3.1. Overview of RNA Sequencing

In this study, ten cDNA libraries were constructed using total RNA purified from the crop tissues of female White King pigeons. Of these, five were from ‘lactating’ pigeons (T1, T2, T3, T4 and T5) and five were from non-‘lactating’ pigeons (C1, C2, C3, C4 and C5). The transcriptome sequencing of the ten libraries generated a total of 1.291 billion raw reads. About 16.44–18.00 gigabase (Gb) of clean bases was acquired for each library and the average clean reads rate was greater than 90%. More than 86% of the total clean reads were mapped onto the rock pigeon colLiv2 reference genome using HISAT2, and 39,289 assembled transcripts were produced. The sequencing, filtering and mapping statistics are shown in Table 1. The distribution of the differentially expressed genes is shown in a volcano plot (Figure 1).

### 3.2. Identification and Characterization of lncRNAs

Of the 39,289 assembled transcripts, 15,138 (38.5%) were predicted to encode proteins and were not used in further analyses (Appendix A). The identified RNAs were not predicted to encode proteins and then they were mapped onto the ALDB database. As a result, 7066 (29.3%) known lncRNAs (Appendix A) were identified while the remaining 17,085 (70.7%) were suggested as novel lncRNAs. Of the novel lncRNAs, 6300 (36.9%) were long intergenic lncRNAs (lincRNAs), 2678 (15.6%) were anti-sense lncRNAs and 8107 (47.5%) were intronic lncRNAs (Appendix A).

The average lengths of the lncRNAs and the mRNAs were 2474 bp and 3549 bp, respectively (Figure 2a). The majority of lncRNAs had two exons (up to 37 exons, an average of 3.1 exons), which was significantly less than that of the mRNAs (up to 290 exons, an average of 24.3 exons) (Figure 2b). A violin plot of the FPKM values for the transcripts indicates that the expression level of the lncRNAs was lower than that of the mRNAs (Figure 2c).

### 3.3. Differential Expression Analysis

The differential expressions of the lncRNAs and mRNAs between the ‘lactating’ crop and non-‘lactating’ crop were analyzed. A total of 6483 differentially expressed mRNAs, 1200 known lncRNAs, and 4966 novel lncRNAs (Appendix A) were identified. The differentially expressed mRNAs and the known and novel lncRNAs are shown in the heatmaps (Figure 3a–c). The differentially expressed lncRNAs gene pairs between ‘lactating’ and non-‘lactating’ crops of the pigeons are shown in Appendix A.

### 3.4. Function Enrichment Related to Pigeon Milk Production

The 6483 differentially expressed genes were significantly enriched in 150 GO terms, including 55 in biological process, 65 in cellular components, and 30 in molecular function (*p* < 0.05) (Appendix A). In the biological process category, cell adhesion, biological adhesion, organonitrogen compound biosynthetic process and small molecule metabolic process were the most significantly enriched terms. In the cellular components, extracellular region part, non-membrane-bound organelle and intracellular non-membrane-bound organelle were the most significant terms. In the molecular function, nucleotide binding, anion binding and small molecule binding were the most significant terms (Figure 4).

The KEGG pathway enrichment analysis revealed that the DEGs were matched to 328 signaling pathways, among which 12 pathways involving 544 genes were significantly enriched (Appendix A). Generally, many genes were significantly enriched in pathways related to the metabolism and biogenesis of macromolecules, genetic information processing and cellular processes. Specifically, RNA transport, ribosomal biogenesis in eukaryotes, amino acids biogenesis and metabolism, carbon metabolism and cell cycle were significantly enriched pathways (Figure 5), and have key roles in milk production, which is consistent with the GO analysis results.

### 3.5. qRT-PCR Verification

Using a qRT-PCR, we validated the reliability of the RNA-seq results by verifying the presence of 11 differentially expressed mRNAs and 11 differentially expressed lncRNAs (Figure 6). The results show that the relative expressions of these genes were consistent with the RNA-seq, which suggests that the RNA-seq data are reliable.

## 4. Discussion

The pigeon crop is a multifunctional organ, as it is used in both food storage and ‘milk’-like substance production during the lactation period. Both female and male pigeons are capable of synthesizing the crop milk. During the reproductive cycle, the crop undergoes dramatic changes in structure, physiology and function. Therefore, crops in different periods are perfect models for analysis of the lactation mechanism. Many previous studies have investigated the expression of genes regulating lactation in pigeons [2,6,7,8,17,18]. However, the molecular mechanism regulating crop milk production in pigeons has not been fully uncovered, perhaps because the biological processes are influenced by several other regulators beside the protein coding genes. In the current study, we characterized the lncRNAs transcriptome of the lactating crops of pigeons by Illumina high-throughput transcriptome sequencing for the first time, in addition to the protein coding transcriptome.

A total of 6483 differentially expressed mRNAs were identified in the lactating and non-lactating crops of pigeons (Appendix A). The up-regulated mRNAs included tryptophan hydroxylase 2 (*TPH2*), which was involved in melatonin and amino acid metabolism, and phosphodiesterase 8B (*PDE8B*) that hydrolyzed the second messenger cAMP, a key regulator of many important physiological processes, including specific signaling in the thyroid gland [19]. Other key genes up-regulated in the ‘lactating’ crop were *STRA6* and *DGAT2*, which were essential for cellular vitamin A uptake and homeostasis and fat pad development and fatty acid homeostasis, respectively [20,21]. *GJA3* facilitated intercellular transport by connecting the cytoplasm of adjacent cells [22].

The identified lncRNAs showed a lower expression, smaller size and fewer exons than the mRNAs’ encoding proteins (Figure 2). This is consistent with the common characteristics of lncRNAs reported in previous studies [23,24]. Only 29.3% of the identified lncRNAs were mapped onto the pigeon lncRNAs in the ALDB database [25,26]. The majority of the identified lncRNAs transcripts were located in the intronic regions of protein coding genes while a minority belonged to the antisense group, at variance with previous reports [27,28]. Among differentially expressed lncRNAs, known lncRNAs were twice as likely to be down-regulated rather than up-regulated, and novel lncRNAs were four times more likely to be down-regulated than up-regulated, suggesting that most of the lncRNAs were down-regulated in the lactating crop. Similar results were reported for lncRNAs in bovine mammary glands compared to other tissues [29].

There were 1200 known and 4966 novel lncRNAs found to be differentially expressed (Appendix A). Many studies have shown that lncRNAs play crucial roles in mammary gland cell proliferation and differentiation, which are important processes in lactation biology. Yu et al. (2017) identified 33 lncRNAs that were differentially expressed in early and late phases of the goat lactation process [30]. Mouse pregnancy-induced non-coding RNA (*mPINC*) and Znfx1 antisense 1 (*Zfas1*) lncRNAs were confirmed to negatively regulate the differentiation of mammary epithelial cells [31]. LncRNAs could regulate neighboring gene expression in cis and affect the expression of a distant gene through the pairing of lncRNAs with mRNA in a trans-acting manner [26]. Thus, these differentially expressed lncRNAs (DE-lncRNAs) might function through targeting mRNA that play an important role in lactation in the pigeon crop. In this study, the MSTRG.102450 lncRNA was predicted to act on the Aldolase C (*ALDOC*) gene and showed opposing expression trends in the lactating and non-lactating crops (Appendix A). *ALDOC* was reported to facilitate the epithelial growth and biosynthesis of milk during lactation and found to be prolactin (PRL)-dependent in the mouse mammary glands [32].

The KEGG pathway analysis revealed that the DE genes were significantly enriched in 12 pathways (Appendix A), mainly categorized into metabolism, genetic information processing and cellular processes. These pathways were related to the metabolism and biosynthesis of carbohydrate, nucleotide, lipid and amino acid (Figure 5). The high activity in these pathways fulfilled the large energy requirement for the epithelial cell proliferation and milk composition synthesis [33,34]. In the biosynthesis of the amino acids pathway, many of the up-regulated genes in the lactation period were related to fatty acid biosynthesis. *PGAM1* was verified to facilitate cis-9, trans-11 conjugated linoleic acid (CLA) synthesis in the bovine mammary gland [35]. CLA, mainly found in milk, has been shown to exert various physiological functions as anticarcinogenic, antidiabetic and antihypertensive, and effectively prevent lifestyle diseases or metabolic syndromes in humans and animals [36]. In the present study, *PGAM1* was 2.5-fold up-regulated in the ‘lactating’ crop. Isocitrate dehydrogenase (*IDH*), the cytosolic form of NADP+, was a primary source of NADPH, which was required for de novo fatty acid synthesis in the lactating bovine mammary gland [37]. *IDH1* and *IDH3B* were significantly up-regulated in the lactating crop, suggesting that the crop of a lactating pigeon synthesizes more fatty acid than the non-lactating crop (Appendix A).

In the propanoate metabolism pathway, the up-regulated genes promoted the synthesis of lipids in the lactating crop. *BCKDH* participated in the branched-chain amino acid catabolic pathway by catalyzing the oxidative decarboxylation of the branched-chainα-keto acids [38]. *ACAT1* and *ACAT2* were acetyl-CoA acetyltransferases, corresponding to two enzymes localized in the mitochondria and cytoplasm respectively. The functions of these two enzymes were catalyzing the formation of acetoacetyl-CoA and maintaining the metabolic balance [39]. *ACACA*, along with *FASN*, were key enzymes of lipid synthesis [40]. Lactation was a dynamic process that involves large scales of cell proliferation, and the cell cycle was the main biological pathway regulating cell proliferation [41].

The cell cycle pathway was significantly enriched and the genes promoting mitosis were pronouncedly up-regulated in the lactating crop. The *BUB* gene family encoded the proteins as part of a large multi-protein kinetochore complex and were proven to be central components of the checkpoint regulatory pathway [42]. The up-regulation of *BUB1B*, *BUB1* and *BUB3* might promote extensive proliferation of the epithelia in the ‘lactating’ crop. The *CDC* family, including the genes *CDC6*, *CDC7*, *CDC16*, *CDC25A*, *CDC25B* and *CDC45*, was up-regulated in the lactating crop (Appendix A). They could disrupt the mitotic integrity and induce the division of the epithelial cell [43].

## 5. Conclusions

These findings contribute to our understanding of the molecular mechanisms involved in pigeon milk production. In this study, DEGs and significantly enriched KEGG pathways were identified, many of which appear to regulate milk production and epithelial tissue proliferation via propanoate/carbon metabolism. Additionally, this study provides fundamental evidence for the understanding of the regulation of lncRNAs in the molecular pathways of lactation in pigeons. To our knowledge, this is the first study to analyze lncRNAs’ functions in the ‘lactating’ crop of pigeon. Further research to verify the functions of the lncRNAs will provide invaluable insight into the molecular mechanisms of pigeon milk production.

## Figures and Tables

**Figure 1 genes-11-00201-f001:**
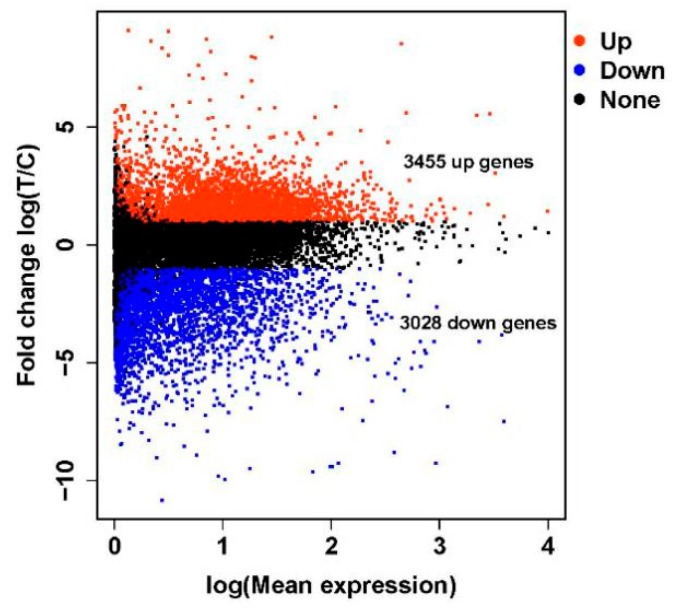
A volcano plot of the differentially expressed genes between ‘lactating’ and non-‘lactating’ crops.

**Figure 2 genes-11-00201-f002:**
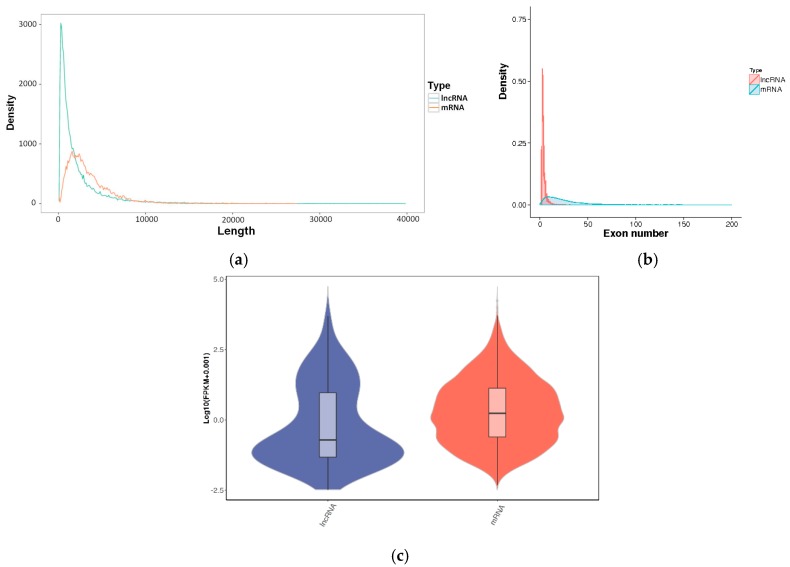
The characteristics of lncRNAs and mRNAs. (**a**) Length distribution; (**b**) Exon number distribution; (**c**) FPKM.

**Figure 3 genes-11-00201-f003:**
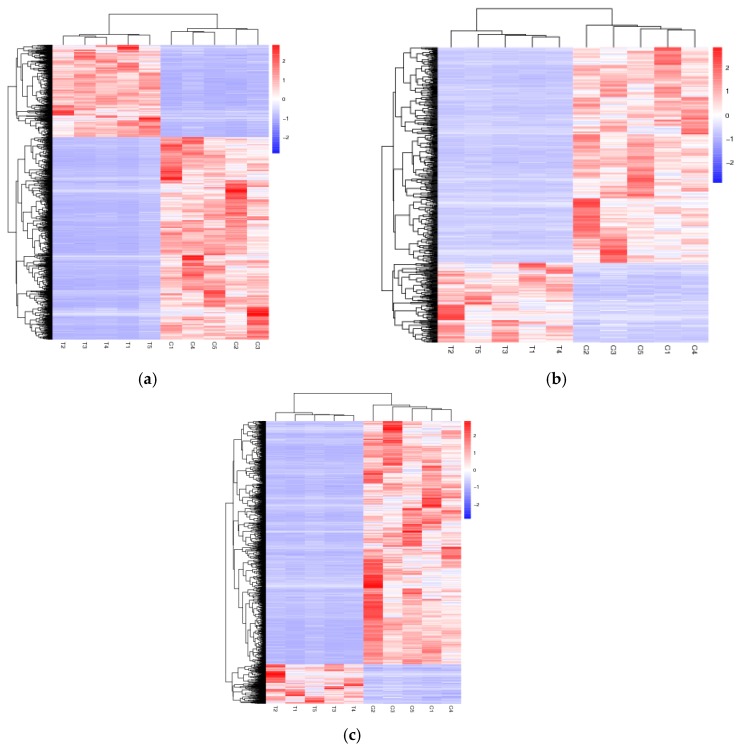
Heat maps of the distinguishable expression profiles between ‘lactating’ and non-‘lactating’ pigeon crops. (**a**) Hierarchical clustering of the differentially expressed mRNAs; (**b**) Hierarchical clustering of the differentially expressed known lncRNAs; (**c**) Hierarchical clustering of the differentially expressed novel lncRNAs. T1, T2, T3, T4 and T5 represent ‘lactating’ pigeons. C1, C2, C3, C4 and C5 represent non-‘lactating’ pigeons.

**Figure 4 genes-11-00201-f004:**
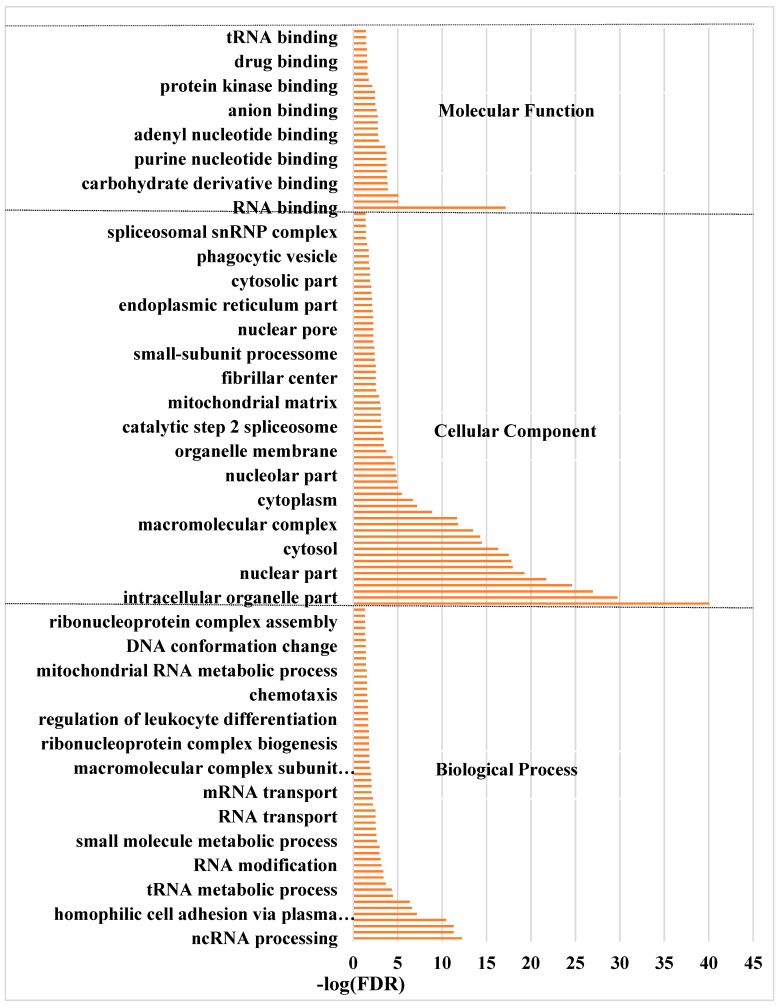
The significantly enriched GO terms of the differentially expressed genes between ‘lactating’ and non-‘lactating’ pigeon crops (FDR < 0.05). BP: biological process; CC: cellular component; MF: molecular function.

**Figure 5 genes-11-00201-f005:**
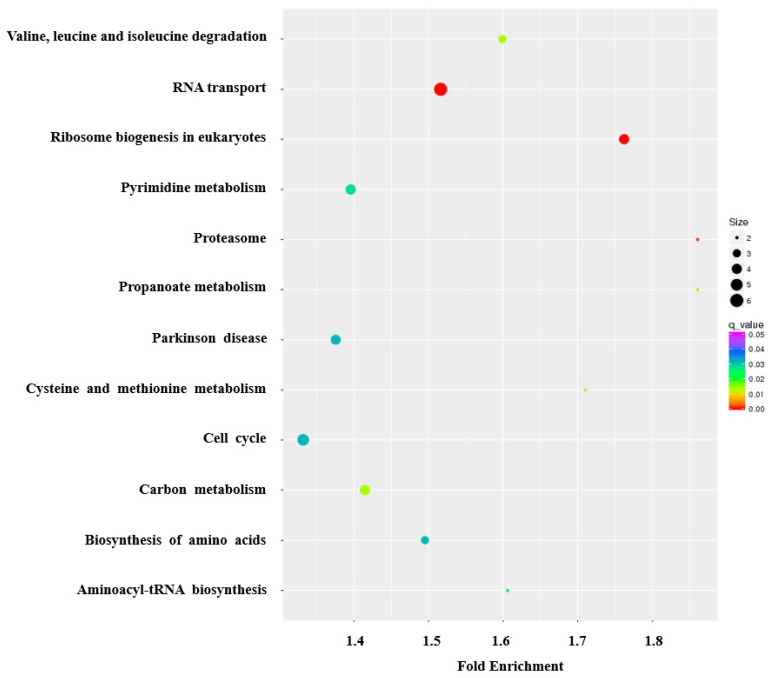
The KEGG pathway analysis of the differentially expressed genes, shown in bubble chart format. The Y-axis label represents significant pathways, and the X-axis label represents fold enrichment. The size of the bubbles represents the levels of differentially expressed genes enriched in the pathway. The color of the bubbles represents significance.

**Figure 6 genes-11-00201-f006:**
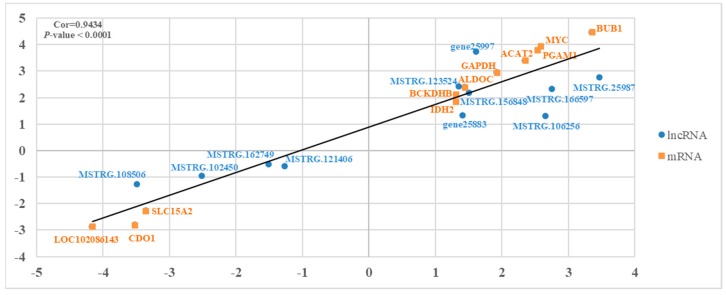
An illustration of the qPCR confirmation for the RNA-seq. The correlations of the gene expression level of the 11 differentially expressed genes and the 11 lncRNAs between lactating and non-‘lactating’ crops using RNA-Seq and a qRT-PCR. The X-axis and Y-axis show the log_2_Foldchange (T/C) measured by the RNA-seq and qRT-PCR, respectively. T represents ‘lactating’ pigeons and C represents non-‘lactating’ pigeons. The β-actin gene was used as a housekeeping internal control.

**Table 1 genes-11-00201-t001:** Output statistics and annotation information of the sequencing reads for each sample.

Sample	Raw Reads	Clean Reads	Raw Bases	Clean Bases	Clean Reads Rate (%)	Clean Q30 Rate (%)	Mapped Reads	Mapping Rate (%)	MultiMap Reads	MultiMap Rate (%)
T1	130,167,700	118,548,640	19,525,155,000	17,782,296,000	91.07	93.46	102,594,735	86.54	1,762,257	1.49
T2	126,086,574	115,719,768	18,912,986,100	17,357,965,200	91.78	93.67	100,909,904	87.20	1,870,468	1.62
T3	129,766,416	118,451,046	19,464,962,400	17,767,656,900	91.28	93.18	100,393,163	84.75	2,107,619	1.78
T4	131,819,754	117,900,806	19,772,963,100	17,685,120,900	89.44	92.68	100,695,085	85.41	1,779,623	1.51
T5	134,001,318	119,376,642	20,100,197,700	17,906,496,300	89.09	93.31	101,578,639	85.09	1,756,397	1.47
C1	123,105,520	109,577,762	18,465,828,000	16,436,664,300	89.01	93.49	95,044,586	86.74	1,440,056	1.31
C2	128,175,504	119,045,316	19,226,325,600	17,856,797,400	92.88	93.80	106,685,232	89.62	1,338,762	1.12
C3	131,962,046	120,026,894	19,794,306,900	18,004,034,100	90.96	93.37	103,784,877	86.47	1,380,399	1.15
C4	125,346,964	111,498,914	18,802,044,600	16,724,837,100	88.95	93.55	95,706,661	85.84	1,420,326	1.27
C5	131,369,282	118,473,848	19,705,392,300	17,771,077,200	90.18	93.45	103,127,477	87.05	1,496,137	1.26

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
