# Peer review of "Analysis of Long Non-Coding RNAs and mRNAs Associated with Lactation in the Crop of Pigeons (Columba livia)"

_genes, 2020, doi:10.3390/genes11020201_

Round 1

Reviewer 1 Report

The authors have mostly addressed my comments. I suggest they change "were showed" on line 165 to "are shown". It would also be helpful to specify whether the lncRNA-gene pairs in Table 2 are trans-targets.

It is still not clear why these 11 pairs were chosen in particular, given that according to Table R2, over 430,000 lncRNA-mRNA pairs were identified with a Correlation coefficient > |0.9|. Are these mRNAs expected to have a particularly important  function in crop milk production? If a clear, logical explanation for highlighting these 11 pairs cannot be given, I recommend the authors remove Table 2 from the main text of the manuscript, and just provide Table R2 as Supplementary Information.

Also, the image of Figure 4 (Figure 3 in the initial submission) does not appear in the file I download, when I try to view it using Adobe Acrobat or Apple Preview. This should be corrected.

Author Response

Response to reviewer 1:

Q1: I suggest they change "were showed" on line 165 to "are shown".

A1: We have changed these words.

Q2: Are these mRNAs expected to have a particularly important function in crop milk production? If a clear, logical explanation for highlighting these 11 pairs cannot be given, I recommend the authors remove Table 2 from the main text of the manuscript, and just provide Table R2 as Supplementary Information.

A2: These mRNAs have important function but they are not the only or special ones playing roles in crop milk production. We take the reviewer’s advice to remove Table 2 and provide as the Supplementary Information.

Q3: Also, the image of Figure 4 (Figure 3 in the initial submission) does not appear in the file I download, when I try to view it using Adobe Acrobat or Apple Preview. This should be corrected.

A3: This has been corrected.

Reviewer 2 Report

Brief summary

This is a novel study that investigates the complex signalling occurring during lactation in the pigeon crop by analysing lncRNA expression and mRNA expression.

Broad comments

There are grammatical and typographical errors in the text. Please proof-read and fix. Gene names need to be in italics, as does “et. al.”.

Specific comments

Introduction

Line 36. Only male emperor penguins lactate.

Line 55. Reference 9 is not the correct reference.

Line 70. It would be great to mention why lncRNAs may be important for future research into pigeon milk production.

Materials and Methods

2.2 Include how the animals were euthanised. CO2 asphyxiation? Anaesthetic overdose?

2.5 Which data repository has the raw data been uploaded to? What were the stringent mapping conditions? Ie. 95/98/100% match? More details needed on methods used for mapping and assembly.

2.6 What were the conditions used for each of the softwares used? It needs to be detailed enough so that it could be repeated.

2.7 Which statistical test was used to determine differential expression? More detail is needed about GO and KEGG analysis ie. Which identifier was used to upload the data into DAVID ie. UniGene, Official Gene ID etc and what was used as the background list (if not Columba livia, was a custom background list used?), how many mapped and how many were mapped to a GO ID? Which conditions were used for the analysis ie FDR or Bonferroni correction?

2.8 Is this quantitative real-time PCR (qPCR) using cDNA or reverse transcription qPCR (RT-PCR) using RNA? Please see the MIQE guidelines: Bustin et. al., The MIQE Guidelines: Minimum Information for Publication of Quantitative Real-Time PCR Experiments, Clinical Chemistry, Volume 55, Issue 4, 1 April 2009, Pages 611–622, https://doi.org/10.1373/clinchem.2008.112797. There is a lot of information missing about your qPCR experiments that needs to go in the materials and methods and supplementary data. Ie. How much RNA was reverse transcribed, if a universal reverse primer was used, what was the sequence (there is only one primer listed in Table S1 for each gene/lncRNA)? Etc.

Results

Line 187. Figure 4 is missing from the manuscript – possibly a formatting issue?

Discussion

Please reference the figure numbers and supplementary table numbers etc in the discussion when referring to expressed genes and lncRNAs.

Author Response

Response to reviewer 2:

Q1: There are grammatical and typographical errors in the text. Please proof-read and fix. Gene names need to be in italics, as does “et. al.”.

A1: All these has been corrected.

Q2: Line 36. Only male emperor penguins lactate.

A2: This has been corrected.

Q3: Line 55. Reference 9 is not the correct reference.

A3: The reference has been changed.

Q4: Line 70. It would be great to mention why lncRNAs may be important for future research into pigeon milk production.

A4: The importance of lncRNAs for future research has been highlighted.

Q5: 2.2 Include how the animals were euthanised. CO2 asphyxiation? Anaesthetic overdose?

A5: The animals were euthanised by CO2 asphyxiation.

Q6: 2.5 Which data repository has the raw data been uploaded to? What were the stringent mapping conditions? Ie. 95/98/100% match? More details needed on methods used for mapping and assembly.

A6: We listed the parameters of HISAT2 and StringTie that we used for mapping and assembly. In this mapping condition, the average mapping rate was 86.47%. And the detailed mapping rates of each samples were described in Table1. The stringent conditions were used to obtain the candidate lncRNAs, and we reexplained in 2.6.

Q7: 2.6 What were the conditions used for each of the softwares used? It needs to be detailed enough so that it could be repeated.

A7: The conditions of the software were added. And the methods were detailed.

Q8: 2.7 Which statistical test was used to determine differential expression? More detail is needed about GO and KEGG analysis ie. Which identifier was used to upload the data into DAVID ie. UniGene, Official Gene ID etc and what was used as the background list (if not Columba livia, was a custom background list used?), how many mapped and how many were mapped to a GO ID? Which conditions were used for the analysis ie FDR or Bonferroni correction?

A8: We used DESeq to determine differential expression. Official Gene ID was used and the background list was Columba livia. The mapping results in GO ID and KEGG pathway were displayed in Supplementary Table S8-Table S11. We used FDR to analyze.

Q9: 2.8 Is this quantitative real-time PCR (qPCR) using cDNA or reverse transcription qPCR (RT-PCR) using RNA? Please see the MIQE guidelines: Bustin et. al., The MIQE Guidelines: Minimum Information for Publication of Quantitative Real-Time PCR Experiments, Clinical Chemistry, Volume 55, Issue 4, 1 April 2009, Pages 611–622, https://doi.org/10.1373/clinchem.2008.112797. There is a lot of information missing about your qPCR experiments that needs to go in the materials and methods and supplementary data. Ie. How much RNA was reverse transcribed, if a universal reverse primer was used, what was the sequence (there is only one primer listed in Table S1 for each gene/lncRNA)? Etc.

A9: The qPCR in this study was performed on cDNA that was reverse-transcribed from RNA. Furthermore, one reviewer suggested to use qRT-PCR to indicate this method. We have checked all the qRT-PCR data in this study according to the MIQE Guidelines and supplemented all the detailed data and information in the revised method item 2.8 qRT-PCR assay. All the primers for each gene/lncRNA are included in Table S1, and every primer was used for the amplification based on the cDNA that was reverse-transcribed from RNA, so one primer for each is correct.

Q10: Line 187. Figure 4 is missing from the manuscript – possibly a formatting issue?

A10: Yes, it is a formatting issue. We have revised it accordingly.

Q11: Please reference the figure numbers and supplementary table numbers etc in the discussion when referring to expressed genes and lncRNAs.

A11: We have referenced the figures and tables in the discussion.

This manuscript is a resubmission of an earlier submission. The following is a list of the peer review reports and author responses from that submission.

Round 1

Reviewer 1 Report

This is a very well performed study investigating the mysterious process of pigeon lactation.  The authors spared no cost performing a very deep sequencing of RNA to identify the key transcriptional factors involved in pigeon lactation.  The authors have only scratched the surface of the data interpretation, this dataset can be downloaded and investigated from many different aspects, with special value due to the depth of sequencing. It is well done to do qPCR confirmation as well.  My minor comments to the authors

make sure that the raw dataset is made public Legend of Figure 3 "All the GO terms showed were significantly enriched" enriched in which group? Be specific through the text regardless if you specified it earlier. remove the text "The Enrichment of Kegg" from the title of figure 4. To be particular - kegg was not enriched, kegg pathways were.

Reviewer 2 Report

In the manuscript “Analysis of long non-coding RNAs and mRNAs associated with lactation in the crop of pigeon (Columba livia), Ma et al. perform a transcriptomic comparison of the crops of lactating vs. non-lactating pigeons. They identify a large number of transcripts that they consider to be differentially-expressed, which they categorize into protein-coding mRNAs, known lncRNAs, and novel lncRNAs. They also perform some preliminary comparisons of expression changes between several lncRNAs and their candidate target genes, with interesting results. Further, they also identify several differentially-expressed genes that lactating pigeons share in common with lactating mammals. Together, their data provide potentially-useful insights into the molecular mechanisms regulating lactation in pigeons. However, there are a number of major and minor problems that need to be addressed.

Major problems:

In the Methods, Section 2.7 states that the significance threshold to categorize a gene as differentially expressed (DE) was “P value < 0.05 and |log2foldchange| >= 1.” These criteria do not adjust for multiple-testing, however, which can lead to a large number of false-positive results. As an illustration of this danger, the authors report that 6,483 (nearly 43%) of the 15,138 mRNAs that were detected were classified as differentially-expressed according to their criteria. This number seems exceptionally high. The authors should adopt the more-stringent criteria Q value < 0.05, which adjusts for multiple testing, to avoid having a large number of false-positives. They also should report this statistic for their DE transcripts in Supplemental Tables S4-S6. This statistic is calculated by the Ballgown R package that the authors use to perform their DE gene analysis.

In relation to Item #1, the authors should revisit their GO-term and KEGG functional enrichment analyses using only the genes that have met the more-stringent threshold.

In Section 3.1, the authors state that “This is the deepest RNA sequencing of pigeon to date.” To help determine whether the authors sequenced at sufficient depth to identify most of the differentially-expressed genes, they should provide a volcano plot that displays the amount of differential-expression of each transcript on one axis, and the average FPKM of the transcript on the other axis. It would also be helpful to indicate what proportion of transcripts in the reference pigeon genome annotation they detected in their analysis.

In Section 3.3, the authors state that “With the potential target gene predicting, 11 garget genes related to 11 lncRNAs were showed in Table 2.” This sentence has some grammatical problems, but more importantly, the authors need to explain why they chose these 11 lncRNA-mRNA pairs in particular. Were these the only pairs identified through their target gene prediction? If not, how many did they identify, and why did they choose to report these 11? Were they the only ones that had a significant correlation coefficient? How did the authors determine whether a particular correlation was statistically significant? The authors should also clarify the criteria they used to identify candidate target genes in Section 2.7: is it simply any gene that is within 100 kb of each lncRNA?

In the Figure 4 legend, the authors state that the size of each dot is related to the “amount of differentially expressed genes enriched in the pathway.” It is unclear what the authors mean by “amount”; are they referring to the number of DE genes? If so, why does Proteasome have a size of 2, while in Supplementary Table S10, it states that 28 upregulated genes map to the KEGG pathway of “Proteasome?” The authors should clarify what “amount” really means.

If the authors adopted a threshold of |log2foldchange| >= 1 to classify a DE transcript, why do so many transcripts in Figure 5 have a value less than this according to the x-axis?

In Discussion, lines 63-64, the authors report that MSTRG.102450 (down) and its putative target gene ALDOC (up) showed opposite changes in expression. This is consistent with the data reported in Table 2, but in Figure 5, both ALDOC and MSTRG.102450 are shown to be upregulated, both in the RNA-seq and in the qRT-PCR data. The authors need to correct this discrepancy.

Minor problems:

Everywhere (ie. Materials and Methods, Results section 3.5, Figure 5, Supplementary Table S1) the authors refer to “qPCR,” they should change it to “qRT-PCR” to indicate that they are performing qPCR on cDNA that was reverse-transcribed from RNA. Page 1, “the genal differences” should be changed to “changes in gene expression”. Page 2, in the sentence beginning “Recently, Holt et al.” the authors conclude with “total length of 14.3 Mb”. I’m assuming that the authors are referring to the N50 value of the scaffolds in this assembly, not the total size of the genome assembly? This should be clarified. In the very next sentence, the authors cite references 11-13, which have nothing to do with analyzing pigeon transcriptomes via microarray. These citations should be removed. In Section 2.5, “ploy-N” should be “poly-N,” and “merged in to transcripts” should be “merged into transcripts.” In Section 2.6, the authors should state that the purpose of blasting the candidate lncRNAs to the ALDB database was to classify a lncRNA as “known” vs. “novel.” This appears in 3.2, but it should be in the Methods. In Section 3.2 “had protein coding potency” should be changed to “were predicted to encode proteins,” and “with no protein coding potency” should be changed “were not predicted to encode proteins.” In Figure 1 legend, part c, “Expression feature” should be changed to “Expression level” or simply “FPKM.” In Figure 4, the x-axis should be “Fold Enrichment” rather than “Rich_ratio.” In the discussion, line 50-51 should be re-written as “among differentially-expressed lncRNAs, known lncRNAs were twice as likely to be down-regulated as up-regulated, and novel lncRNAs were 4 times as likely to be down-regulated as up-regulated.” Line 92 should be re-written as “The upregulation of BUB1B, BUB1, and BUB3 might promote” Line 94 should have a comma after “crop” The meaning of Lines 94-95 should be clarified. I’m not sure what the authors mean by “induced epithelial cell division after proliferation by disrupting the mitotic integrity.” Line 97 should be re-written as “contribute to our understanding” Line 98 should be rewritten as “production. DEGs and significantly enriched KEGG pathways have been identified” The Acknowledgement statement is a fragment. It should conclude with a phrase like “technical expertise” or “technical assistance,” etc.